# Ionizing Radiation Activates Mitochondrial Function in Osteoclasts and Causes Bone Loss in Young Adult Male Mice

**DOI:** 10.3390/ijms23020675

**Published:** 2022-01-08

**Authors:** Kimberly K. Richardson, Wen Ling, Kimberly Krager, Qiang Fu, Stephanie D. Byrum, Rupak Pathak, Nukhet Aykin-Burns, Ha-Neui Kim

**Affiliations:** 1Center for Musculoskeletal Disease Research and Center for Osteoporosis and Metabolic Bone Diseases, Division of Endocrinology and Metabolism, Department of Internal Medicine, University of Arkansas for Medical Sciences, Little Rock, AR 72205, USA; KKRichardson@uams.edu (K.K.R.); WLing@uams.edu (W.L.); QFu@uams.edu (Q.F.); 2Division of Radiation Health, Department of Pharmaceutical Sciences, University of Arkansas for Medical Sciences, Little Rock, AR 72205, USA; KJKrager@uams.edu (K.K.); RPathak@uams.edu (R.P.); NAykinburns@uams.edu (N.A.-B.); 3Department of Biochemistry and Molecular Biology, University of Arkansas for Medical Sciences, Little Rock, AR 72205, USA; SBYRUM@uams.edu

**Keywords:** ionizing radiation, bone loss, bone resorption, osteoclast, mitochondria, Sirtuin-3

## Abstract

The damaging effects of ionizing radiation (IR) on bone mass are well-documented in mice and humans and are most likely due to increased osteoclast number and function. However, the mechanisms leading to inappropriate increases in osteoclastic bone resorption are only partially understood. Here, we show that exposure to multiple fractions of low-doses (10 fractions of 0.4 Gy total body irradiation [TBI]/week, i.e., fractionated exposure) and/or a single exposure to the same total dose of 4 Gy TBI causes a decrease in trabecular, but not cortical, bone mass in young adult male mice. This damaging effect was associated with highly activated bone resorption. Both osteoclast differentiation and maturation increased in cultures of bone marrow-derived macrophages from mice exposed to either fractionated or singular TBI. IR also increased the expression and enzymatic activity of mitochondrial deacetylase Sirtuin-3 (Sirt3)—an essential protein for osteoclast mitochondrial activity and bone resorption in the development of osteoporosis. Osteoclast progenitors lacking Sirt3 exposed to IR exhibited impaired resorptive activity. Taken together, targeting impairment of osteoclast mitochondrial activity could be a novel therapeutic strategy for IR-induced bone loss, and Sirt3 is likely a major mediator of this effect.

## 1. Introduction

Human exposure to ionizing radiation (IR) comes from man-made sources that range from nuclear power generation to medical uses of radiation for diagnosis or treatment. Reduction in bone mineral density, bone growth retardation in childhood, and spontaneous fracture in postmenopausal women are well-established deleterious effects of IR on bone health [1,2,3,4,5,6]. IR primarily exerts its deleterious effects by damaging nuclear DNA [7]. This damage to DNA can cause mitochondrial dysfunction and result in cell death, carcinogenesis, or tissue degeneration [7]. DNA damage-induced mitochondrial dysfunction leads to activation or inactivation of a number of proteins, resulting in widespread downstream changes in cellular pathways [7,8,9,10,11,12]. Among them, signaling from the nucleus to mitochondria is less well-studied than other pathways; therefore, a complete understanding of the deleterious effects of IR exposure may help combat the resultant diseases, including bone disorders.

Balance between the function of bone-resorbing osteoclasts and bone-building osteoblasts is required for maintaining bone metabolism. However, pathological or physiological changes, such as IR exposure, inflammation, or aging, increase the number of osteoclasts and decrease the number of osteoblasts, tilting the balance in favor of bone resorption and eventually causing bone diseases [13,14,15,16,17,18,19,20,21,22]. The damaging effects of IR on bone mass have been established in both mice and humans [17,18,23,24,25,26,27,28,29,30,31,32]. Although the cellular alterations in the skeleton as a response to IR vary depending on the dose of IR and method of radiation delivery, the bone loss most likely results from an increase in osteoclast number and bone resorption [17,18,23,24,33,34,35,36,37]. Human biomarker studies have shown that, after IR exposure, damage to bone structure is related to increased osteoclast function [38,39,40]. Interestingly, in mice, osteoclast number and activity increase early after exposure to IR, but no significant change occurs in the number of osteoblasts or in markers of bone formation [17,18,32,41,42]. Therefore, osteoclasts may be primary mediators of the IR-induced bone loss.

Osteoclasts are uniquely capable of resorbing bone matrix by virtue of their ability to secrete lysosomal enzymes into a sealed microenvironment that tightly adheres to the bone area targeted for removal [43]. Likely due to the high energy demands of this task, a distinct feature of osteoclasts is the high abundance of mitochondria [44,45]. Sirtuin-3 (Sirt3) is the primary protein deacetylase in mitochondria and plays a crucial role in mitochondrial quality control, which is affected in age-related and metabolic diseases [46,47,48,49]. Much of the current knowledge on Sirt3 function comes from studies using Sirt3-knockout mice under pathological conditions such as aging, IR exposure, and oxidative stress. In general, Sirt3 plays a protective role in these conditions, most likely by promoting deacetylation of its target proteins. However, it is important to note that Sirt3 does not play a major role under non-stressed conditions [46,47,48,49]. We and others have shown that genetic ablation or pharmacological inhibition of Sirt3 in mice attenuates age- and estrogen deficiency-associated bone loss [50,51,52]. This effect was associated with decreased osteoclast mitochondrial-specific autophagy, called mitophagy, and impaired bone resorption [52]. More recently, we also showed that Sirt3 mRNA expression increased significantly in osteoclasts exposed to high linear energy-transfer radiation (a major component of galactic cosmic rays) [33]. Furthermore, altered mitochondrial metabolism in osteoclasts contributes to the loss of bone mass with IR exposure [33]. Collectively, these findings strongly suggest that Sirt3 might be responding to IR-induced stress signals but, in turn, over-activating the formation and activity of osteoclasts; however, there is currently little to no direct evidence that an increase in Sirt3 expression/activation is related to the IR-induced increase in osteoclast number and bone resorption or that these adverse effects result from enhanced mitochondrial function in osteoclast-lineage cells. We show herein that exposure to multiple fractions or singular exposure to the same total dose of total body irradiation (TBI) activates Sirt3 in osteoclasts and induces trabecular bone loss in adult mice and that deletion of Sirt3 prevents the increased bone resorption activity caused by IR exposure.

## 2. Results

### 2.1. Fractionated Exposure or a Single Exposure to the Same Total Radiation Dose Causes Trabecular Bone Loss in Young Adult Male Mice

To determine the effects of radiation delivery modes on bone damage, mice were exposed to either 10 fractions of 0.4 Gy TBI/week or (fractionated exposure; Figure 1A) or a single fraction of the same total dose of TBI (1 fraction of 4 Gy TBI; or singular exposure Figure 1H). As determined by micro-CT, trabecular bone volume (bone volume per tissue volume [BV/TV]) and 3D bone mineral density (BMD) decreased at both the spine (Figure 1) and the femur (Figure 2) after both TBI exposure. This decrease was associated with a decrease in trabecular number and an increase in trabecular spacing, while no changes were detected in trabecular thickness of the spine (Figure 1). The singular exposure caused a decrease in trabecular number and an increase in trabecular spacing of the femur, while it had no effect on trabecular thickness, as seen in the spine (Figure 2J–L). The trabecular microarchitecture in the fractionated exposure group trended lower bone mass than in the control group (Figure 2D–F). In contrast to trabecular bone mass, no difference in femoral (Figure 3) and spinal (data not shown) cortical thickness was observed following fractionated and singular exposure in young adult males. In addition, exposure to 0.5 Gy singular exposure failed to alter trabecular bone mass in 3-month-old male C57BL/6 mice (Appendix A). This is consistent with a previous report that a similar dose (0.1 Gy) of γ-radiation had no effect on bone mass in young male C57BL/6 mice [21]. Taken together, these results indicate that a certain level of IR induces bone loss in young adult mice and that distinct mechanisms are responsible for tissue damage in trabecular versus cortical bone.

### 2.2. Fractionated Exposure or a Single Exposure to the Same Total Radiation Dose Induces Osteoclastic Bone Resorption-Driven High Bone Turnover in Young Adult Male Mice

IR exposure-related low bone mass and strength are associated with increased osteoclast numbers and decreased osteoblast numbers [16,17,18,20,21,22,24,32]. To elucidate the cellular mechanisms by which the fractionated and singular exposure induce trabecular bone loss, we performed dynamic histomorphometric analysis of the trabecular surface of the spine. Fractionated exposure had no effect on osteoclast or osteoblast number (Figure 4A–D). Strikingly, however, the serum level of C-terminal telopeptide of type 1 collagen (CTx), a bone resorption marker, was dramatically higher in both TBI-treated mice than in control mice (Figure 4E,F), consistent with low bone mass. Surprisingly, TBI caused an approximate 60% increase in bone formation rate (BFR) and mineralized surfaces (MS), but it had no effect on mineral apposition rate (MAR) (Figure 4G–J). These findings strongly suggest that the low bone mass of the irradiated mice was, at least in part, due to an increase in osteoclast activation and high bone turnover.

### 2.3. Fractionated and Singular IR Exposure Promotes Osteoclast Maturation

Based on the results of the experiment described in Figure 4, we proceeded to investigate the mechanism of action by which fractionated and singular IR exposure increased osteoclast activation in osteoclast progenitors. In these experiments, we first cultured bone marrow-derived macrophages (BMMs) from mice with fractionated IR exposure and sham controls in the presence of M-CSF and RANKL. Cultures of BMMs from the IR group formed significantly more osteoclasts (>10 nuclei) than those from control mice (Figure 5A,B). Importantly, the mature osteoclasts from the IR group were much larger than those from the control group (Figure 5C). To examine the effects of singular IR exposure on osteoclast maturation, we subjected pre-osteoclasts to two different doses of singular exposure (2 and 4 Gy). Increased osteoclast size and spreading were observed following singular exposure (Figure 5D–F).

### 2.4. IR Increases Sirt3 Activity and Mitochondrial Respiration in Osteoclasts

It has been reported that mitochondrial Sirt3 contributes to bone loss caused by aging or estrogen deficiency, and this effect is associated with impaired osteoclast maturation and resorption activity [50,51,52]. More recently, we found that cultured osteoclasts from mice exposed to galactic cosmic rays have significantly higher Sirt3 mRNA levels [33]. Therefore, we next examined whether fractionated radiation also altered Sirt3-mediated mitochondrial function in osteoclasts. Fractionated radiation significantly increased Sirt3 mRNA levels and enzymatic activity in osteoclasts (Figure 6A,C). We did not observe any change in c-Fos mRNA levels, which is known to play a crucial role in early osteoclast differentiation (Figure 6A). However, the expression of late/terminal osteoclast differentiation markers was increased in whole spinal bones from irradiated mice (Figure 6B). Importantly, we also found that after IR exposure, osteoclast progenitors from young Sirt3-knockout mice exhibited impaired resorptive activity when cultured in the presence of RANKL (Figure 6D,E). The inhibitory effect of Sirt3 deletion on bone resorbing activities seen in cells from irradiated mice (−74.25%; Figure 6E) was mild in cells from control mice (−27.25%; Figure 6E). These results demonstrate that IR causes bone loss in young adult mice by increasing osteoclast differentiation and function, and Sirt3 might be responsible for this effect. 

We used mass spectrometry to analyze the global proteome of osteoclasts from mice exposed to fractionated exposure and received no radiation. Gene Ontology enrichment analysis confirmed that several mitochondrial pathways were upregulated in BMMs from the IR group (not shown). However, the overall fold-change in proteins was minimal between treatments (Appendix A). Interestingly, we identified a slight decrease in mitochondrial quality control protein Pex6 levels in osteoclasts from mice exposed to fractionated IR (Appendix A).

Because of the key role of Sirt3 in osteoclast mitochondrial activity, we examined whether fractionated exposure alters cellular and mitochondrial bioenergetics in osteoclasts by performing Seahorse extracellular flux analysis. IR significantly increased the basal mitochondrial oxygen consumption rate (Figure 7A) and ATP-linked respiration (Figure 7B) in osteoclasts from irradiated mice, suggesting there was an increased energy demand in osteoclasts. At the same time, osteoclasts exposed to IR exhibited a significant increase in oxygen consumption rate associated with proton leak, meaning that IR induced a disruption of electron flow through the electron transport chain (Figure 7C). IR increased the maximum respiration (maximal electron transport chain activity) and nonmitochondrial respiration (Figure 7D,E). Despite the increased maximum respiration in IR group, the similar reserved respiratory capacities (difference between maximum and basal oxygen consumption rate) observed in sham and radiated groups implied that the cells were operating close to their bioenergetic limit following radiation exposure (Figure 7F). Taken together, these results suggest that increased osteoclast mitochondrial activity is a major cause of IR-induced bone loss in young adult mice.

## 3. Discussion

The high risk of fractures related to IR-induced bone loss represents a major clinical problem for both women and men [1,2,3,4,5,6]. While the molecular mechanisms underlying the activation of osteoclasts or inactivation of osteoblasts after exposure to IR remain unknown, both IR-induced inflammation and mitochondrial dysregulation are thought to be the likely culprits. Thus, in this study, we examined the skeletal effects of TBI delivered as multiple fractions or as a single exposure to the same total dose using a Cs^137^ γ-rays source on various bone parameters. Our data indicate that in adult male mice, Sirt3 activation is essential for IR-induced bone loss, and these effects most likely result from highly activated mitochondrial metabolism in late-stage osteoclastogenesis.

Because the damaging effects of IR are thought to be caused by an increase in osteoclast number and bone resorption [17,18,41], our observation of increased bone formation caused by IR was somewhat unexpected. However, similar findings have been reported for mice irradiated under several other conditions of γ-radiation. For example, TBI (2 Gy) of young male C57BL/6 mice decreased the trabecular bone volume fraction in the proximal tibiae and lumbar vertebrae, as seen in the current study, but increased MAR [30]. Likewise, mice exposed to 6 Gy γ-radiation had a transient increase in BFR due, at least in part, to the activation of post-mitotic osteoblast-lineage cells [53], while mice exposed to 2 Gy X-rays had a rapid increase in MAR, but a subsequent decline in BFR within 2 to 3 weeks after the IR [23]. Therefore, this range of IR dose (at least not high-dose) TBI may cause time-dependent high bone turnover, as indicated by an observed increase in BFR and/or MAR, as well as enhanced bone resorption activity. We and others have reported that estrogen deficiency—one of the most common causes of osteoporosis—increases osteoclastic bone resorption associated with high bone turnover and causes trabecular bone loss in humans and mice [13,54,55,56,57,58,59], demonstrating that an increase in osteoclast number and function is a major contributor to the development of osteoporosis. 

We have previously shown that IR exposure causes DNA damage and senescence in cultured osteoblasts, similar to the effects seen in osteoprogenitors from old mice [16]. In that study, we subjected cultured osteoblasts to 10 Gy of γ-radiation. Likewise, higher doses of IR than 4 Gy cause DNA damage and apoptosis of osteoblast-lineage cells, including osteoblast progenitors, bone lining cells, and osteocytes in young adult mice [22,60,61,62]. Chandra et al. reported that mice exposed to focal radiation have a reduction in osteoblast activation and bone formation, as indicated by an observed decrease in calcein double-labeling on the trabecular bone surface or the levels of serum bone formation markers [22,61,62]. Interestingly, a single high dose of focal radiation caused a time-dependent reduction in osteoclast number and activity [22,61,62]. Overall, these findings support the notion that the cellular alterations in the bone as a response to IR vary depending on the dose of IR and method of radiation delivery. 

The current study demonstrate that trabecular bone mass and number significantly decreased, while no change occurred in the cortical bones following IR exposure. Similar reduction in trabecular, but not in cortical, bone mass was observed in our previous study where mice were exposed to particle radiation—a type of IR that deposit greater energy per unit distance than γ-rays [33], suggesting that IR of different qualities exert similar adverse effects on bone mass. Future studies are needed to further dissect the pathways by which IR impacts osteoclastogenesis and bone resorption and why these predominate in distinct bone compartments.

While our results point to an increase in bone resorption as the cause of decreased trabecular bone mass in mice exposed to IR, a disproportionate effect on bone resorption could mask an effect of IR on bone formation. Mice exposed to 4 Gy of γ-radiation exhibited a highly activated BFR on the trabecular bone surface that was not explained by changes in osteoblast numbers. One possible explanation for the increase in bone formation could be an increase in growth factors or cytokines released from the matrix during bone resorption. In view of the major increase in BFR in irradiated mice, it is also possible that this IR exposure condition plays a key role in supporting osteoblasts by stimulating osteoclast activation to express coupling factors. Nonetheless, further biochemical and genetic studies are necessary to elucidate this possibility. Importantly, we used only male mice in the current study without addressing any possible sex differences. This must be taken into consideration while interpreting our data. Indeed, female and male C57BL/6 mice exhibit differences in bone aging—another feature of osteoporotic pathogenesis [63]. 

We found here that IR exposure increased Sirt3 mRNA levels and enzymatic activity, which in turn stimulated osteoclast maturation and bone resorption, most likely by promoting mitochondrial metabolism. In the absence of Sirt3, the increased bone resorptive activity caused by IR was completely prevented, as seen in aged Sirt3-knockout mice [52]. We performed a global proteomic analysis in osteoclasts from mice exposed to IR or sham controls. Gene Ontology enrichment analysis showed that several mitochondrial pathways were upregulated in BMMs from the irradiated mice. Nevertheless, the overall fold-change in proteins was minimal between treatments, suggesting post-translational modification such as acetylation, rather than protein levels, may be responsible for the Sirt3-mediated changes in mitochondrial function in osteoclasts from irradiated mice. Interestingly, IR results in a slight decrease in Pex6 protein levels in osteoclasts. Pex6 is known to suppress aging defects in mitochondria and is implicated in the control of both mitochondrial bioenergetics and structure by regulating ATP synthase activity [64]. Pex6 also plays a key role in the autophagy of peroxisomes (pexophagy) [65]. Given the major role of Sirt3 in osteoclast mitochondrial activity and bone resorption in the development of osteoporosis, it is also possible that Sirt3 regulates osteoclast mitophagy or other mitochondrial quality control processes via Pex6 in highly activated osteoclasts in the context of IR exposure. Therefore, further genetic studies will be required to elucidate whether Sirt3 and target proteins such as Pex6 in osteoclasts, osteoblasts, or other cell types have any impact on IR-mediated bone loss. 

Taken together, we provide evidence of possible cellular and molecular mechanisms by which IR causes bone loss in young adults. In addition to age- or sex steroid deficiency-associated bone loss, Sirt3 might also contribute to increased bone resorption in the pathogenesis of osteoporosis caused by IR exposure. While our study has limitations, including lack of a female group and inconsistent age of mice in fractionated and singular IR exposure, the evidence we provide here demonstrates the importance of the regulation of osteoclast mitochondria and its contribution to skeletal damage. Therefore, our study may advance the field by providing fundamental knowledge to inform optimal therapeutic approaches for the treatment of bone disorders in patents, including elderly veterans, current military service members, and astronauts who might be exposed to multiple fractions of IR.

## 4. Materials and Methods

### 4.1. Animals and Low-Dose IR Exposure

C57BL/6J (also known as B6) male mice were obtained from the Jackson Laboratory (Bar Harbor, ME, USA) and housed 4 to 5 animals per cage at the University of Arkansas for Medical Sciences (UAMS). The mice received standard rodent chow and water ad libitum until they reached 2 or 6 months of age. The mice were exposed to low doses (Sham and 4 Gy) of IR (J. L. Shepherd Mark 1 Model 68A 137Cs γ-irradiator) at a dose rate of 1 Gy/min for 10 weeks. TBI schedules included the following plans: (i) single TBI exposures: one fraction of 4 Gy exposure to 2-month-old mice for 8 months and (ii) fractionated TBI exposures: 40 cGy (0.4 Gy) fractions once per week for 10 weeks (total dose 4 Gy) to 6-month-old mice. Sham-irradiated mice were treated in the same way but did not receive IR. The skeletal phenotype was characterized with micro-CT, strength measurements, and histomorphometry.

### 4.2. Micro-CT Analysis

Left femurs and lumbar vertebra (L5) from irradiated or sham control mice were dissected, cleaned by removing adherent tissues, fixed in Millonig’s phosphate buffer (Leica Microsystems, Buffalo Grove, IL, USA), and stored in 100% ethanol. Bones were scanned at 12 μm isotropic voxel size, 500 projections (medium resolution, E = 55 kVp, I = 72 μA, 4 W, integration time 150 ms and threshold 200 mg/cm^3^), integrated into 3-D voxel images (1024 × 1024 pixel matrices for each individual planar stack) from the distal epiphysis to the mid-diaphysis to obtain a number of slices variable between 650 and 690 [63]. Nomenclature conforms to recommendations of the American Society for Bone and Mineral Research. Cortical thickness was determined at the diaphysis (18 slices, midpoint of the bone length as determined in scout view) at a threshold of 200 mg/cm^3^. Two-dimensional evaluation of trabecular bone was performed on contours of the cross sectional acquired images excluding the primary spongiosa and cortex. Contours were drawn starting 8–10 slices away from the growth plate from the distal metaphysis to the diaphysis of the femur to obtain 151 slices (12 μm/slice). The L5 was scanned from the rostral growth plate to the caudal growth plate to obtain 233 slices. BV/TV in the vertebra was determined using 100 slices (1.2 mm) of the anterior vertebral body immediately inferior to the superior growth plate. For all trabecular bone measurements contours were drawn every 10 to 20 slices. Voxel counting was used for bone volume per tissue volume measurements and sphere filling distance transformation indices were used for trabecular microarchitecture with a threshold value of 220 mg/cm^3^, without pre-assumptions about the bone shape as a rod or plate.

### 4.3. Dynamic Histomorphometric Analysis

We used the terminology recommended by the Histomorphometry Nomenclature Committee of the American Society for Bone and Mineral Research. After μCT analysis, the left femurs from irradiated or sham control mice were embedded undecalcified in methyl methacrylate. Calcein labels and osteoclasts were quantified on trabecular bone surfaces in 5-µm-thick longitudinal sections using an Olympus BX53 microscope and Olympus DP73 camera (Olympus America, Inc., Waltham, MA, USA) interfaced with a digitizer tablet with OsteomeasureTM software version 4.1.0.2 (OsteoMetrics Inc., Decatur, GA, USA). The osteoclasts were stained with naphthol AS-MX and Fast Red TR salt (Sigma-Aldrich, St. Louis, MO, USA). Sections were also stained with 0.3% toluidine blue in phosphate-buffered citrate, pH 3.7, to visualize osteoblasts, osteoid, and cement lines. The following primary measurements were made: bone surface (BS), single label surface (sL.S), double label surface (dL.S), inter-label thickness (Ir.L.Th), osteoclast number (N.Oc, /μm), and osteoclast surface (Oc.S, %). The following derived indices were calculated: mineralizing surface (MS, %), mineral apposition rate (MAR, μm/d), and bone formation rate (BFR, μm^3^/μm^2^/d). One section per sample was analyzed by a histopathologist blinded to the study groups.

### 4.4. CTx and Osteocalcin ELISA

Blood was collected into 1.7-mL EDTA-coated microcentrifuge tubes by retro-orbital bleeding. Blood was then kept on ice for 1 h and centrifuged at 6150 *g* at 4 °C for 10 min to separate serum from cells before analysis. Circulating C-terminal telopeptide of type 1 collagen (CTx) and osteocalcin in serum were measured using a mouse RatLaps (CTx-I) ELISA kit (Immunodiagnostic Systems, Boldon, United Kingdom) and Osteocalcin enzyme immunoassay kit (Thermo Fisher, Waltham, MA, USA) according to the manufacturer’s instructions.

### 4.5. Osteoclast Differentiation

BMMs were obtained as described previously [66]. Briefly, whole bone marrow cells were cultured with 10% FBS, 1% PSG, 10 ng/mL of M-CSF (R&D Systems, Minneapolis, MN, USA) for 12 h. Non-adherent bone marrow cells were then replated in Petri dishes with 10 ng/mL M-CSF and adherent BMMs were harvested as osteoclast precursors 4–5 days later. To generate pre-osteoclasts or mature osteoclasts, BMMs were cultured with 30 ng/mL M-CSF and 30 ng/mL RANKL (R&D Systems) for 3 or 5 days at 37 °C. To enumerate osteoclasts, the cells were stained with tartrate-resistant acid phosphatase (TRAP) using a Leukocyte Acid Phosphatase Assay Kit following the manufacturer’s instructions (Sigma-Aldrich). A giant osteoclast formation was identified as a multinucleated (>10 nuclei) TRAP-positive cell.

### 4.6. Quantitative RT-PCR

Total RNA was purified from cultured osteoclasts or bone tissues using TRIzol re-agent (Thermo Fisher). RNA was quantified using a Nanodrop instrument (Thermo Fisher) and 1–2 μg of total RNA was used to synthesize cDNA using a High-Capacity cDNA Reverse Transcription kit (Applied Biosystems, Foster City, CA, USA) according to the manufacturer’s instructions. Transcript abundance in the cDNA was measured by quantitative PCR using TaqMan Universal PCR Master Mix (Thermo Fisher). The primers and probes for murine Sirt3 (Mm00452131_m1), c-Fos (Mm00487425_m1), Acp5 (Mm00432448_m1), and Ctsk (Mm00484039_m1) were manufactured by the TaqMan^®^ Gene Expression Assays service (Applied Biosystems). Relative mRNA expression levels were normalized to the housekeeping gene ribosomal protein S2 (Mm00475528_m1) and mRNA fold-change for various genes was determined by the ∆Ct method.

### 4.7. Sirt3 Activity Assay

BMMs were isolated as described above and stimulated with RANKL to form preosteoclasts on 6-well plates. Cell pellets were washed with cold PBS and treated with mitochondria isolation reagents supplied in the kit. The cell lysates were then kept on ice for 5 min and centrifuged at 700 *g* for 10 min at 4 °C. The supernatant was transferred to new tubes and centrifuged at 12,000 *g* for 5 min at 4 °C to isolate the mitochondrial potion, after adding 2% CHAPS in Tris to the pellets. The protein concentration of cell lysates was determined using a DC Protein Assay kit (Bio-Rad). Sirt3 activity was measured using a Sirt3 Activity Assay Kit (Abcam, Cambridge, United Kingdom) according to the manufacturer’s directions. The released fluorescent signal was measured in a microplate fluorescence reader with excitation/emission wavelengths of 340/440 nm.

### 4.8. Bone Resorption Assay

BMMs were isolated from 6-month-old female Sirt3-knockout mice and WT littermate controls as previously described [52] and stimulated with RANKL to generate mature osteoclasts on Osteo Assay Surface 24-well plates (Corning Life Sciences, New York, NY, USA) coated with an inorganic bone biomaterial surface. The osteoclasts were removed using a 2% hypochlorite solution for 5 min, washed with distilled water, and dried at room temperature. For Von Kossa staining, plates were treated in darkness with 150 μL/well of 5% (*w*/*v*) aqueous silver nitrate solution for 25 min. Plates were then washed for 5 min with distilled water and incubated in darkness with 150 μL/well of 5% (*w*/*v*) sodium carbonate in a 10% formalin solution. Plates were then washed twice with PBS, rinsed with distilled water, and dried in a 50 °C oven for 25 min. Three wells per group were assessed microscopically. In this assay, the resorbed areas appear white, and the unresorbed mineralized surface appears black.

### 4.9. Mitochondrial Respiration and Cellular Bioenergetics

The BMMs were plated in Seahorse XF96 plates with 30 ng/mL M-CSF for 12 h and treated with 30 ng/mL RANKL for 3 days to generate pre-osteoclasts. The media in the wells was replaced with XF assay media (Agilent, Lexington, MA, USA), and the plates were kept in a non-CO_2_ incubator for 20 min at 37 °C. Total 3 cellular respiration measurements were recorded with the Seahorse XF96 analyzer (Agilent) before 10 μg/mL oligomycin (Sigma-Aldrich) was added to inhibit mitochondrial ATP synthase and measure the decrease in the oxygen consumption rate that is linked to ATP turnover. An oxidative phosphorylation uncoupler, FCCP (10 μM, Sigma-Aldrich), was used to determine the maximal respiration potential of the pre-osteoclasts. The amount of nonmitochondrial oxygen consumption was assessed by inhibiting the electron respiratory chain activity using an antimycin A (10 μM, Sigma Aldrich) and rotenone cocktail (10 μM, Sigma Aldrich). These data were used to calculate the mitochondrial basal respiration, ATP-linked respiration, reserve respiratory capacity, and proton leak, as we described previously [52]. Briefly, we defined the basal mitochondrial respiration as the oxygen consumption rate (OCR) attributed only to the electron transport chain (ETC) activity. Once ETC activity was entirely inhibited by a rotenone/antimycin A cocktail, the remaining OCR was noted as the non-mitochondrial respiration, which represents the cytosolic oxygen consuming enzyme activity. The oxygen consumption rate that was suppressed by oligomycin, ATP synthase (complex V) inhibitor, was reported as the ATP-linked respiration; while we described the residual mitochondrial basal respiration that was not inhibited by the oligomycin as the “proton leak”. The difference between maximum OCR (induced by the uncoupler, carbonyl cyanide 4-(trifluoromethoxy)phenylhydrazone, FCCP) and basal respiration was also calculated and disclosed as the “reserve capacity”.

### 4.10. Preparation of Global Proteome

Purified proteins were reduced, alkylated, and digested using filter-aided sample preparation [67]. Tryptic peptides were labeled using tandem mass-tag isobaric labeling reagents (Thermo Fisher) following the manufacturer’s instructions and combined into 2 multiplex sample groups with a common reference sample. The labeled peptide multiplexes were separated into 36 fractions on a 100 × 1.0 mm Acquity BEH C18 column (Waters, Milford, MA, USA) using an UltiMate 3000 UHPLC system (Thermo Fisher) with a 40 min gradient from 99:1 to 60:40 buffer A:B ratio under basic pH conditions, and then consolidated into 12 super-fractions. Each super-fraction was then further separated by reversed-phase XSelect CSH C18 2.5 µm resin (Waters) on an in-line 150 × 0.075 mm column, using an UltiMate 3000 RSLCnano system (Thermo Fisher). Peptides were eluted using a 75 min gradient from 98:2 to 60:40 buffer A (0.1% formic acid, 0.5% acetonitrile) to B (0.1% formic acid, 99.9% acetonitrile) ratio. Both buffers were adjusted to pH 10 with ammonium hydroxide for offline separation. Eluted peptides were ionized by electrospray (2.2 kV), followed by mass spectrometry (MS) analysis on an Orbitrap Fusion Lumos mass spectrometer (Thermo Fisher) using multi-notch MS3 parameters. MS data were acquired using the FTMS analyzer in top-speed profile mode at a resolution of 120,000 over a range of 375 to 1500 *m*/*z*. Following CID activation with normalized collision energy of 35.0, MS/MS data were acquired using the ion trap analyzer in centroid mode and normal mass range. Using synchronous precursor selection, up to 10 MS/MS precursors were selected for HCD activation with normalized collision energy of 65.0, followed by acquisition of MS3 reporter ion data using the FTMS analyzer in profile mode at a resolution of 50,000 over a range of 100–500 *m*/*z*.

### 4.11. Proteomic Data Analysis

Proteins were identified and reporter ions quantified by searching the Mus musculus database using MaxQuant (Max Planck Institute, Planegg, Germany) with a parent ion tolerance of 3 ppm, a fragment ion tolerance of 0.5 Da, and a reporter ion tolerance of 0.001 Da. Protein identifications were accepted if they could be established with less than 1.0% false discovery and contained at least 2 identified peptides. Protein probabilities were assigned by the Protein Prophet algorithm [68]. TMT MS3 reporter ion intensity values were log2 transformed and missing values were imputed by a normal distribution for each sample using Perseus (Max Planck Institute). TMT batch effects were removed using ComBat [69] to correct for technical variation due to multiplexing samples across multiple TMT10plex batches. Statistical analysis was performed using Linear Models for Microarray Data (limma) with empirical Bayes (eBayes) smoothing to the standard errors [70]. Proteins with an FDR adjusted *p*-value < 0.05 and a fold change >2 were considered significant.

### 4.12. Statistics

All data were analyzed using GraphPad Prism 9 (GraphPad Software, La Jolla, CA, USA). For all graphs, data are represented as the mean ± SD unless otherwise specified. For comparison of two groups, data were analyzed using a two-tailed Student’s *t*-test, and for comparison of 3 groups, one-way ANOVA with Tukey’s post hoc test was used. *p* < 0.05 was considered significant.

## 5. Conclusions

Our data reveal that fractionated or singular TBI exposure causes trabecular, but not cortical, bone loss; increases osteoclast maturation and bone resorption; upregulates Sirt3 expression/activation in osteoclasts and elevates osteoclast mitochondrial activity in mice. Moreover, IR fails to induce bone resorption in osteoclasts lacking Sirt3, suggesting Sirt3 could be a novel therapeutic target to prevent TBI-induced bone damage.

## Figures and Tables

**Figure 1 ijms-23-00675-f001:**
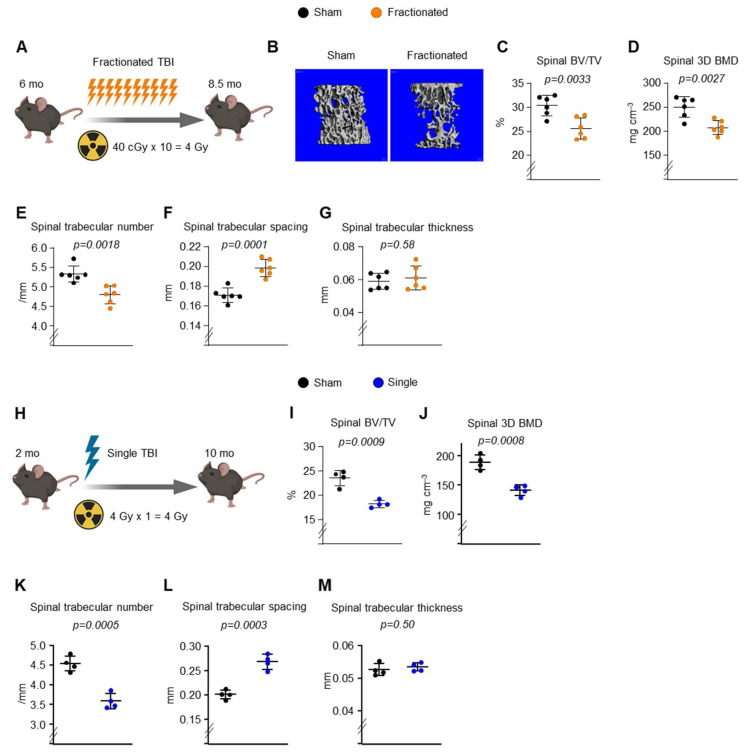
Loss of trabecular bone mass in spine following fractionated and singular exposure. (**A**,**H**) Schematic representation of the scheduling of TBI exposures. (**B**–**G**,**I**–**M**) Imaging and quantification of spinal bones from sham and IR-exposed male mice by micro-CT after sacrifice (*n* = 4–6 animals/group). (**B**) Representative 3D trabecular bone images of spine from 8.5-month-old male C57BL/6 mice. (**C**,**I**) Bone volume over tissue volume (BV/TV), (**D**,**J**) bone mineral density (BMD), (**E**,**K**) trabecular number, (**F**,**L**) spacing, and (**G**,**M**) thickness of trabecular bone measured in the spine of irradiated mice and sham controls by µCT. Data are presented as Mean ± SD. *p* values were determined using Student’s *t* test.

**Figure 2 ijms-23-00675-f002:**
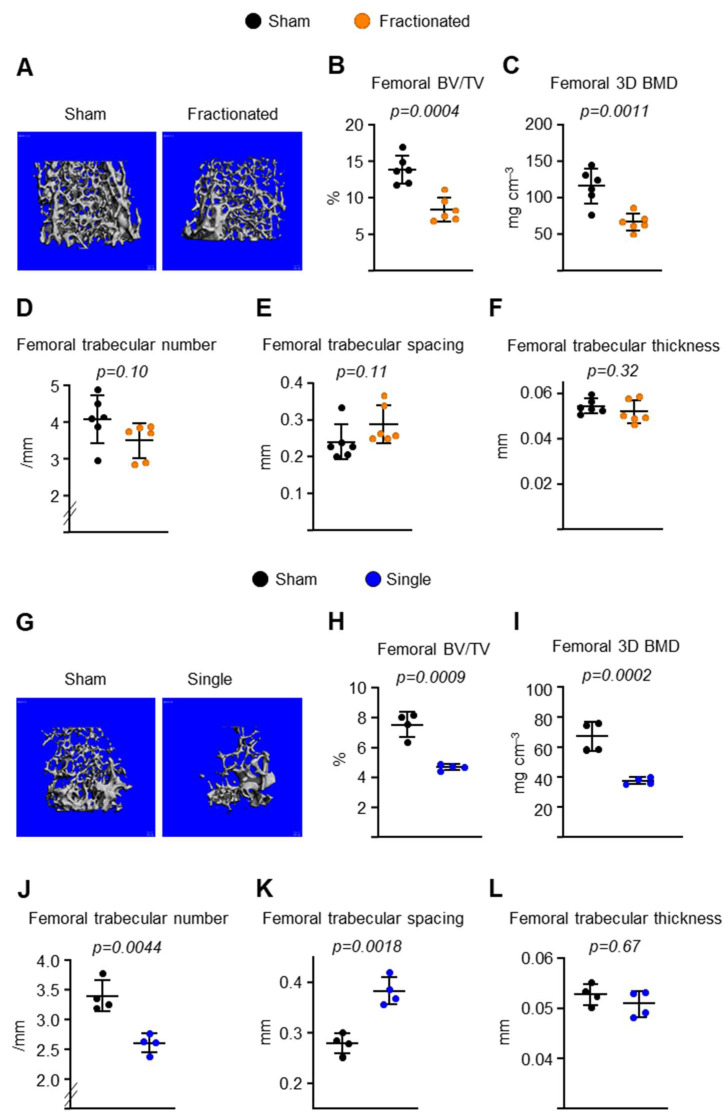
Loss of trabecular bone mass in femur following fractionated and singular TBI exposure. Imaging and quantification of femoral bones from sham and IR-exposed male mice by micro-CT after sacrifice (*n* = 4–6 animals/group). (**A**,**G**) Representative 3D trabecular bone images of femur from (**A**) 8.5- and (**G**) 10-month-old male C57BL/6 mice. (**B**,**H**) Bone volume over tissue volume (BV/TV), (**C**,**I**) bone mineral density (BMD), (**D**,**J**) trabecular number, (**E**,**K**) spacing, and (**F**,**L**) thickness of trabecular bone measured in the femur of irradiated mice and sham controls by µCT. Data are presented as ±SD. *p* values were determined using Student’s *t* test.

**Figure 3 ijms-23-00675-f003:**
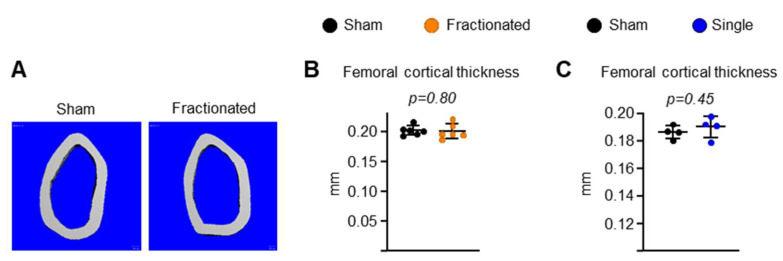
IR does not alter cortical bone mass in young adult male mice. (**A**) Representative 3D cortical bone images of femora from 8.5-month-old male C57BL/6 mice. (**B**) Cortical thickness in femoral diaphysis from fractionated TBI mice and sham controls measured by µCT. *n* = 6 animals/group. *p* values by Student’s *t*-test. (**C**) Cortical thickness in femoral diaphysis from single TBI mice and sham controls measured by µCT. *n* = 4 animals/group. Data are presented as ±SD. *p* values were determined using Student’s *t* test.

**Figure 4 ijms-23-00675-f004:**
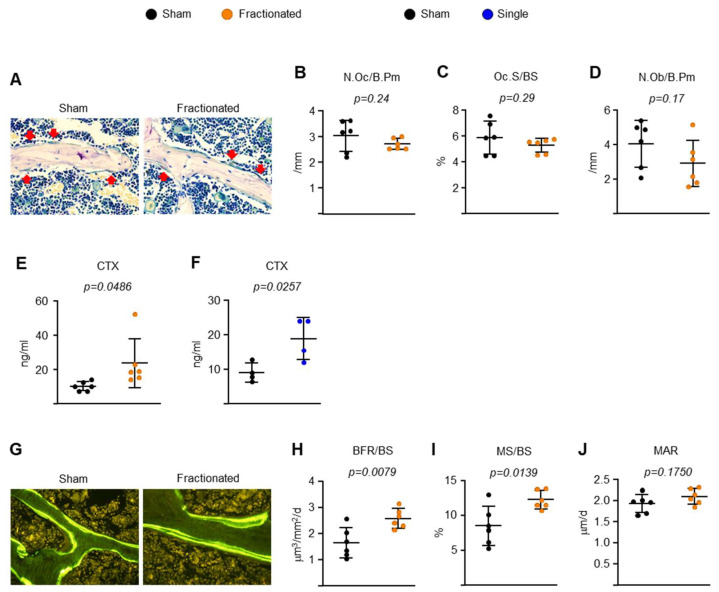
IR increases bone resorption in young adult male mice. (**A**) Representative photomicrographs of non-decalcified spine sections stained for TRAPase activity (red) from 8.5-month-old male C57BL/6 mice. (**B**) Number of osteoclast (N.Oc/B.Pm), (**C**) osteoclast surface, and (**D**) osteoblast (N.Ob/B.Pm) per trabecular bone surface (*n* = 6 animals/group). (**E**,**F**) Serum concentration of a collagen degradation product (CTx) in (**E**) 8.5- and (**F**) 10-month-old male C57BL/6 mice by ELISA (*n* = 4–6 animals/group). (**G**) Representative photomicrographs of trabecular bone surface labeled with tetracycline (fluorescent green) in undecalcified spine sections from 8.5-month-old male C57BL/6 mice. (**H**) Bone formation rate (BFR/BS), (**I**) mineralizing surface (MS/BS), and (**J**) mineral apposition rate (MAR) determined by tetracycline labels (*n* = 6 mice/group, one section per mouse was analyzed). Data are presented as ±SD. *p* values were determined using Student’s *t* test.

**Figure 5 ijms-23-00675-f005:**
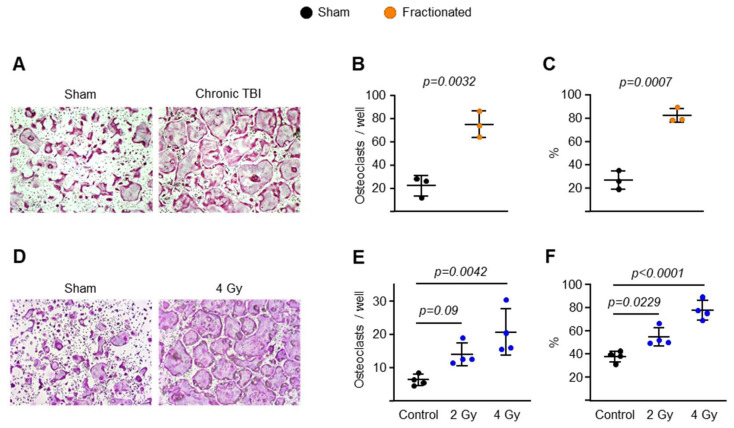
IR activates late-stage osteoclastogenesis. (**A**–**C**) BMMs were isolated from 8.5-month-old male C57BL/6 sham- or IR-exposed mice 10 weeks after irradiation and cultured with M-CSF (30 ng/mL) and RANKL (30 ng/mL) for 4 days. (**A**) Representative pictures, (**B**) number and (**C**) total area of TRAP-positive multinucleated osteoclasts (>10 nuclei) generated from BMMs. (**D**–**F**) BMMs were isolated from 6-month-old male C57BL/6 mice and cultured with M-CSF (30 ng/mL) and RANKL (30 ng/mL) for 4 days. γ-irradiation was subjected to pre-osteoclasts after 2-day cultures with the indicated doses. (**D**) Representative pictures, (**E**) number and (**F**) total area of TRAP-positive multinucleated osteoclasts (>10 nuclei) generated from BMMs. Data are presented as ±SD. *p* values were determined using Student’s *t* test (**B**,**C**) or 1-way ANOVA (**E**,**F**). All measures were performed in cultured BMMs pooled from 3 mice/group.

**Figure 6 ijms-23-00675-f006:**
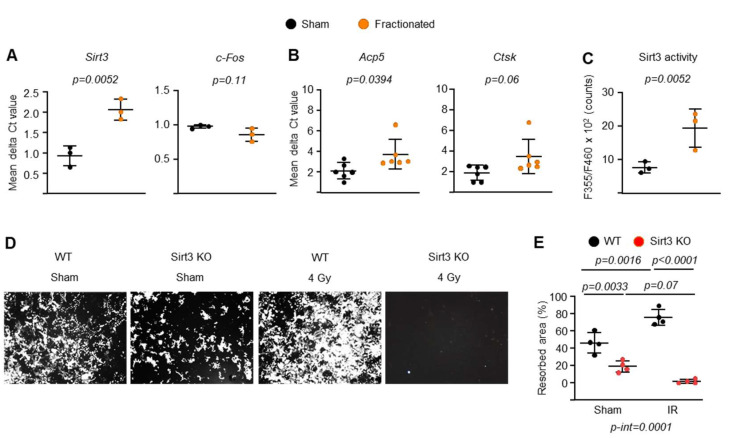
IR activates mitochondrial Sirt3 in osteoclasts. (**A**,**C**) BMMs were isolated from 8.5-month-old male C57BL/6 sham- or IR-exposed mice 10 weeks after irradiation and cultured with M-CSF (30 ng/mL) and RANKL (30 ng/mL) for (**A**) 3 or (**C**) 2 days. (**A**) Sirt3 and c-Fos levels in mRNA of cultured osteoclasts measured by qPCR. (**B**) Osteoclast marker levels in mRNA of lumbar vertebra (L1) measured by qPCR. (**C**) Sirt3 enzymatic activity in cultured osteoclasts measure by fluorescence reader. (**D**,**E**) BMMs were isolated from 6-month-old male Sirt3-knockout mice and WT littermates and cultured with M-CSF (30 ng/mL) and RANKL (30 ng/mL) for 5 days. 4 Gy γ-irradiation was subjected to pre-osteoclasts after 3-day cultures. (**D**) Representative pictures and (**E**) resorbed areas of Von Kossa–stained bone biomaterial surface. The resorbed areas appear white and the unresorbed mineralized surface appears black. Data are presented as ±SD. *p* values were determined using Student’s *t* test (**A**–**C**) or 2-way ANOVA (**E**). All measures were performed in cultured BMMs pooled from 3 mice/group. Interaction terms generated by 2-way ANOVA are shown below the graph.

**Figure 7 ijms-23-00675-f007:**
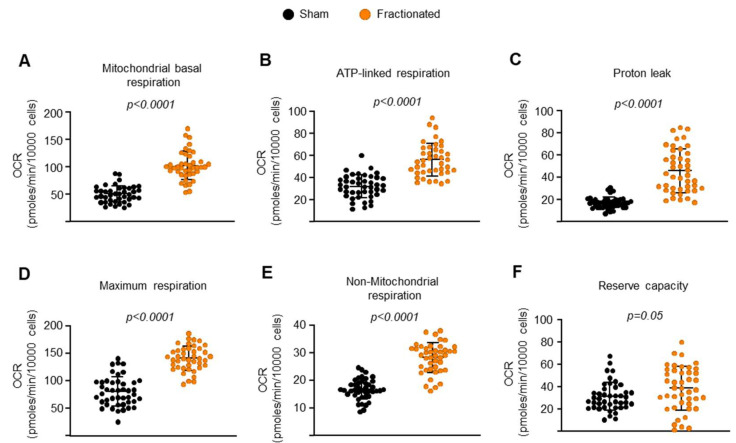
IR increases overall mitochondrial and non-mitochondrial respiration. (**A**–**F**) BMMs were isolated from 8.5-month-old male C57BL/6 sham- or IR-exposed mice 10 weeks after irradiation and cultured with M-CSF (30 ng/mL) and RANKL (30 ng/mL) for 3 days. (**A**–**D**,**F**) Different fractions of mitochondrial and (**E**) nonmitochondrial respirations per cell, in osteoclasts, measured by Seahorse (*n* = 41–43 wells/group). Data are presented as ±SD. *p* values were determined using Student’s *t* test.

## Data Availability

Not applicable.

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
