# Peer review of "Ionizing Radiation Activates Mitochondrial Function in Osteoclasts and Causes Bone Loss in Young Adult Male Mice"

_ijms, 2022, doi:10.3390/ijms23020675_

Round 1

Reviewer 1 Report

In this study, Richardson et al. addressed the cellular and molecular mechanisms underlying damaging effects of ionizing radiation on bone mass. The authors showed that fractionated exposure and/or same total dose single exposure resulted in decreased trabecular bone mass but not cortical bone mass in young male mice. In the irradiated mice, the levels of bone resorption marker CTX were increased, and bone formation rate was increased. The authors also revealed that irradiated mice-derived BMMs generated significantly higher number and bigger size of osteoclasts than control mice-derived BMMs. Additionally, the authors revealed that irradiation induced expression of Sirt3, a critical protein for osteoclast mitochondrial activity. Irradiation increased osteoclastic bone resorbing activities, which was completely diminished in Sirt3-deficient cells. Finally, the authors showed irradiation increased overall mitochondrial and non-mitochondrial respiration. Take together, the authors suggested that increased mitochondrial activity is a major cause of irradiation-induced bone loss.

This is an interesting study. This study will contribute to progress of this research field. I would like to make several comments to this study.

  1. According to the authors previous report (JCI insight doi: 10.1172/jci.insight.146728), Sirt3-deficient cells show impaired bone resorbing activities in aged mice regardless irradiation. What about the basic (non-irradiated) bone resorbing activities in 6-month-old male Sirt3-deficient mice-derived osteoclasts (Figure 6D, E)? Do Sirt3-deficinent osteoclasts show the same level of bone resorbing activities with/out irradiation? Or does it change?

  1. It would be better to show mitochondrial respiration values in Sirt3-deficinet cells with/out irradiation (Figure 7), so that relevance between irradiation-induced mitochondrial activities and Sirt3 become clear.

  1. Please describe a little more detail about how to measure the mitochondrial respiration values (basal respiration, ATP-linked respiration, proton leak, maximum respiration, non-mitochondrial respiration and reverse capacity) in figure legend or in materials and methods. This will help readers to understand the data.

Author Response

In this study, Richardson et al. addressed the cellular and molecular mechanisms underlying damaging effects of ionizing radiation on bone mass. The authors showed that fractionated exposure and/or same total dose single exposure resulted in decreased trabecular bone mass but not cortical bone mass in young male mice. In the irradiated mice, the levels of bone resorption marker CTX were increased, and bone formation rate was increased. The authors also revealed that irradiated mice-derived BMMs generated significantly higher number and bigger size of osteoclasts than control mice-derived BMMs. Additionally, the authors revealed that irradiation induced expression of Sirt3, a critical protein for osteoclast mitochondrial activity. Irradiation increased osteoclastic bone resorbing activities, which was completely diminished in Sirt3-deficient cells. Finally, the authors showed irradiation increased overall mitochondrial and non-mitochondrial respiration. Take together, the authors suggested that increased mitochondrial activity is a major cause of irradiation-induced bone loss.

This is an interesting study. This study will contribute to progress of this research field. I would like to make several comments to this study.

According to the authors previous report (JCI insight doi: 10.1172/jci.insight.146728), Sirt3-deficient cells show impaired bone resorbing activities in aged mice regardless irradiation. What about the basic (non-irradiated) bone resorbing activities in 6-month-old male Sirt3-deficient mice-derived osteoclasts (Figure 6D, E)? Do Sirt3-deficinent osteoclasts show the same level of bone resorbing activities with/out irradiation? Or does it change?

In the revised manuscript we show that the inhibitory effect of Sirt3 deletion on bone resorbing activities seen in cells from irradiated mice (–74.25%; Figure 6E) was mild in cells from control mice (–27.25%; Figure 6E). These results suggest that Sirt3 contributes to the increase in osteoclast function and bone resorption during IR exposure.

It would be better to show mitochondrial respiration values in Sirt3-deficinet cells with/out irradiation (Figure 7), so that relevance between irradiation-induced mitochondrial activities and Sirt3 become clear.

The reviewer is correct. Unfortunately, however, we do not have enough Sirt3-deficient BMMs to repeat Seahorse extracellular flux analysis and we cannot generate enough cells to provide such data in the time allotted for the revised manuscript submission. In a separate line of work, we are examining the skeleton and bone cells of irradiated Sirt3 KO mice. We hope to further dissect the relevance between irradiation-induced mitochondrial activities and Sirt3 in this ongoing study.

Please describe a little more detail about how to measure the mitochondrial respiration values (basal respiration, ATP-linked respiration, proton leak, maximum respiration, non-mitochondrial respiration and reverse capacity) in figure legend or in materials and methods. This will help readers to understand the data.

The following text was included in the methods section in the revised text:

We defined the basal mitochondrial respiration as the oxygen consumption rate (OCR) attributed only to the electron transport chain (ETC) activity. Once ETC activity was entirely inhibited by a rotenone/antimycin A cocktail, the remaining OCR was noted as the non-mitochondrial respiration, which represents the cytosolic oxygen consuming enzyme activity. The oxygen consumption rate that was suppressed by oligomycin,  ATP synthase (complex V) inhibitor, was reported as the ATP-linked respiration; while we described the residual mitochondrial basal respiration that was not inhibited by the oligomycin as the “proton leak”. The difference between maximum OCR (induced by the uncoupler, carbonyl cyanide 4-(trifluoromethoxy)phenylhydrazone, FCCP) and basal respiration was also calculated and disclosed as the “reserve capacity”.

Reviewer #2

This manuscript provides data that Sirt3 regulates osteoclast development and activity in response to ionizing radiation in vivo and in vitro. Data include microCT of bones from irradiated and non-irradiated wild type male mice, and bone marrow cell cultures of differentiating osteoclasts, and serum marker measurements. Data pointed to the main effect of radiation was to stimulate resorption rather than inhibit bone formation, and published studies led to the hypothesis that Sirt3 could be involved in the resorptive phenotype.  Bone resorption assays were performed in one experiment from wild type and irradiated bone marrow cells from wild type and knockout mice. Seahorse assays of radiation treated and control osteoclasts pointed to mitochondrial defects in irradiated cells as expected.  Overall, the study is interesting with many strengths. However, a few deficiencies are noted below for the authors to consider.

Sirt3 knockout osteoclasts were shown only in one assay which lacked a control of knockout cells without radiation treatment. Presumably these cells without radiation would also be unable to resorb bone.

In the revised manuscript we show that the inhibitory effect of Sirt3 deletion on bone resorbing activities seen in cells from irradiated mice (–74.25%; Figure 6E) was mild in cells from control mice (–27.25%; Figure 6E). These results suggest that Sirt3 contributes to the increase in osteoclast function and bone resorption during IR exposure.

Data showing osteoclast differentiation as in Figure 5 in the presence and absence of radiation treatment, or of osteoclasts differentiated from radiation-treated Sirt3 knockout mice is needed to support the conclusion that Sirt3 knockout mice exhibit the expected osteoclast phenotype. Similarly, Seahorse assays from cells derived from SIrt3 knockout mice would be interesting and would further support the conclusions made.

The reviewer raises a very important issue. However, we would mainly focus on the effect of IR exposure here in the current study. Moreover, we do not have enough Sirt3-deficient BMMs to repeat all the assays suggested in the time allotted for the revised manuscript submission. In a separate line of work, we are examining the skeleton and bone cells of irradiated Sirt3 KO mice. We hope to further dissect the relevance between irradiation-induced mitochondrial activities and Sirt3 in this ongoing study.

It appears that only male mice were used in this study. Since sex-dependent differences in bone biology are common, the manuscript needs to emphasize that only male mice were studied, and that effects documented here in males may or may not be different in females.

We agree with the reviewer’s argument and have modified the sentence accordingly.

All figure legends should specify the sex of animals used in the respective data set.

Following the reviewer’s suggestion, we have modified the sentence.

qPCR data should be presented as deltaCT, not delta-deltaCT in scatter plots. This gives a better representation of both the magnitude and relative changes of the mRNAs measured.

We have modified the figures as advised.

Reviewer 2 Report

This manuscript provides data that Sirt3 regulates osteoclast development and activity in response to ionizing radiation in vivo and in vitro. Data include microCT of bones from irradiated and non-irradiated wild type male mice, and bone marrow cell cultures of differentiating osteoclasts, and serum marker measurements. Data pointed to the main effect of radiation was to stimulate resorption rather than inhibit bone formation, and published studies led to the hypothesis that Sirt3 could be involved in the resorptive phenotype.  Bone resorption assays were performed in one experiment from wild type and irradiated bone marrow cells from wild type and knockout mice. Seahorse assays of radiation treated and control osteoclasts pointed to mitochondrial defects in irradiated cells as expected.  Overall, the study is interesting with many strengths. However, a few deficiencies are noted below for the authors to consider.

  1. Sirt3 knockout osteoclasts were shown only in one assay which lacked a control of knockout cells without radiation treatment. Presumably these cells without radiation would also be unable to resorb bone. Data showing osteoclast differentiation as in Figure 5 in the presence and absence of radiation treatment, or of osteoclasts differentiated from radiation-treated Sirt3 knockout mice is needed to support the conclusion that Sirt3 knockout mice exhibit the expected osteoclast phenotype. Similarly, Seahorse assays from cells derived from SIrt3 knockout mice would be interesting and would further support the conclusions made.
  2. It appears that only male mice were used in this study. Since sex-dependent differences in bone biology are common, the manuscript needs to emphasize that only male mice were studied, and that effects documented here in males may or may not be different in females. All figure legends should specify the sex of animals used in the respective data set.
  3. qPCR data should be presented as deltaCT, not delta-deltaCT in scatter plots. This gives a better representation of both the magnitude and relative changes of the mRNAs measured.

Author Response

Reviewer #1

In this study, Richardson et al. addressed the cellular and molecular mechanisms underlying damaging effects of ionizing radiation on bone mass. The authors showed that fractionated exposure and/or same total dose single exposure resulted in decreased trabecular bone mass but not cortical bone mass in young male mice. In the irradiated mice, the levels of bone resorption marker CTX were increased, and bone formation rate was increased. The authors also revealed that irradiated mice-derived BMMs generated significantly higher number and bigger size of osteoclasts than control mice-derived BMMs. Additionally, the authors revealed that irradiation induced expression of Sirt3, a critical protein for osteoclast mitochondrial activity. Irradiation increased osteoclastic bone resorbing activities, which was completely diminished in Sirt3-deficient cells. Finally, the authors showed irradiation increased overall mitochondrial and non-mitochondrial respiration. Take together, the authors suggested that increased mitochondrial activity is a major cause of irradiation-induced bone loss.

This is an interesting study. This study will contribute to progress of this research field. I would like to make several comments to this study.

According to the authors previous report (JCI insight doi: 10.1172/jci.insight.146728), Sirt3-deficient cells show impaired bone resorbing activities in aged mice regardless irradiation. What about the basic (non-irradiated) bone resorbing activities in 6-month-old male Sirt3-deficient mice-derived osteoclasts (Figure 6D, E)? Do Sirt3-deficinent osteoclasts show the same level of bone resorbing activities with/out irradiation? Or does it change?

In the revised manuscript we show that the inhibitory effect of Sirt3 deletion on bone resorbing activities seen in cells from irradiated mice (–74.25%; Figure 6E) was mild in cells from control mice (–27.25%; Figure 6E). These results suggest that Sirt3 contributes to the increase in osteoclast function and bone resorption during IR exposure.

It would be better to show mitochondrial respiration values in Sirt3-deficinet cells with/out irradiation (Figure 7), so that relevance between irradiation-induced mitochondrial activities and Sirt3 become clear.

The reviewer is correct. Unfortunately, however, we do not have enough Sirt3-deficient BMMs to repeat Seahorse extracellular flux analysis and we cannot generate enough cells to provide such data in the time allotted for the revised manuscript submission. In a separate line of work, we are examining the skeleton and bone cells of irradiated Sirt3 KO mice. We hope to further dissect the relevance between irradiation-induced mitochondrial activities and Sirt3 in this ongoing study.

Please describe a little more detail about how to measure the mitochondrial respiration values (basal respiration, ATP-linked respiration, proton leak, maximum respiration, non-mitochondrial respiration and reverse capacity) in figure legend or in materials and methods. This will help readers to understand the data.

The following text was included in the methods section in the revised text:

We defined the basal mitochondrial respiration as the oxygen consumption rate (OCR) attributed only to the electron transport chain (ETC) activity. Once ETC activity was entirely inhibited by a rotenone/antimycin A cocktail, the remaining OCR was noted as the non-mitochondrial respiration, which represents the cytosolic oxygen consuming enzyme activity. The oxygen consumption rate that was suppressed by oligomycin,  ATP synthase (complex V) inhibitor, was reported as the ATP-linked respiration; while we described the residual mitochondrial basal respiration that was not inhibited by the oligomycin as the “proton leak”. The difference between maximum OCR (induced by the uncoupler, carbonyl cyanide 4-(trifluoromethoxy)phenylhydrazone, FCCP) and basal respiration was also calculated and disclosed as the “reserve capacity”.

Reviewer #2

This manuscript provides data that Sirt3 regulates osteoclast development and activity in response to ionizing radiation in vivo and in vitro. Data include microCT of bones from irradiated and non-irradiated wild type male mice, and bone marrow cell cultures of differentiating osteoclasts, and serum marker measurements. Data pointed to the main effect of radiation was to stimulate resorption rather than inhibit bone formation, and published studies led to the hypothesis that Sirt3 could be involved in the resorptive phenotype.  Bone resorption assays were performed in one experiment from wild type and irradiated bone marrow cells from wild type and knockout mice. Seahorse assays of radiation treated and control osteoclasts pointed to mitochondrial defects in irradiated cells as expected.  Overall, the study is interesting with many strengths. However, a few deficiencies are noted below for the authors to consider.

Sirt3 knockout osteoclasts were shown only in one assay which lacked a control of knockout cells without radiation treatment. Presumably these cells without radiation would also be unable to resorb bone.

In the revised manuscript we show that the inhibitory effect of Sirt3 deletion on bone resorbing activities seen in cells from irradiated mice (–74.25%; Figure 6E) was mild in cells from control mice (–27.25%; Figure 6E). These results suggest that Sirt3 contributes to the increase in osteoclast function and bone resorption during IR exposure.

Data showing osteoclast differentiation as in Figure 5 in the presence and absence of radiation treatment, or of osteoclasts differentiated from radiation-treated Sirt3 knockout mice is needed to support the conclusion that Sirt3 knockout mice exhibit the expected osteoclast phenotype. Similarly, Seahorse assays from cells derived from SIrt3 knockout mice would be interesting and would further support the conclusions made.

The reviewer raises a very important issue. However, we would mainly focus on the effect of IR exposure here in the current study. Moreover, we do not have enough Sirt3-deficient BMMs to repeat all the assays suggested in the time allotted for the revised manuscript submission. In a separate line of work, we are examining the skeleton and bone cells of irradiated Sirt3 KO mice. We hope to further dissect the relevance between irradiation-induced mitochondrial activities and Sirt3 in this ongoing study.

It appears that only male mice were used in this study. Since sex-dependent differences in bone biology are common, the manuscript needs to emphasize that only male mice were studied, and that effects documented here in males may or may not be different in females.

We agree with the reviewer’s argument and have modified the sentence accordingly.

All figure legends should specify the sex of animals used in the respective data set.

Following the reviewer’s suggestion, we have modified the sentence.

qPCR data should be presented as deltaCT, not delta-deltaCT in scatter plots. This gives a better representation of both the magnitude and relative changes of the mRNAs measured.

We have modified the figures as advised.

Round 2

Reviewer 2 Report

The authors have addressed the critiques very well. This will be of interest to readers.